# Investigating the correlation between candidate teachers' acceptance of generative artificial intelligence and artificial intelligence literacy across various disciplines

Berker Kurt[1], Gözdegül Arık Karamık[2]*, Ali Özkaya[3]

**1** Department of Turkish Language and Social Sciences Education, Faculty of Education, Akdeniz University, Antalya, Türkiye, **2** Department of Mathematics and Science Education, Faculty of Education, Akdeniz University, Antalya, Türkiye, **3** Department of Mathematics and Science Education, Faculty of Education, Akdeniz University, Antalya, Türkiye

* gkaramik@akdeniz.edu.tr

## Abstract

This study examines Generative Artificial Intelligence (GenAI) acceptance and Artificial Intelligence Literacy (AIL) levels among prospective teachers, using variables for comparative analysis and identifying influencing factors. The research uses an explanatory sequential mixed methods approach. Quantitative data were obtained from 723 prospective teachers and qualitative data from 48 prospective teachers. Data collection included an Information Form, GenAI Acceptance Scale, and AIL Scale for quantitative data, with interview forms for qualitative data. Parametric tests, independent samples t-test, ANOVA, and Pearson correlation analyzed quantitative data, while factors influencing GenAI acceptance and AIL were identified through themes using MAXQDA. Acceptance levels showed no significant differences by gender or daily internet use; however, differences emerged regarding department, grade level, AI tools used, and self-perceived proficiency. AIL showed significant differences in gender, department, grade, tool usage, and proficiency level, with higher scores among those trained in artificial intelligence. Qualitative data clarify the quantitative findings. Factors affecting GenAI acceptance include daily use, problem-solving, learning applications, mentor usage, assistance from others, proficiency, productivity, discipline-specific skills, and task efficiency. Factors influencing AIL include understanding AI importance, ethical considerations, AI support in daily life, explaining AI, understanding deep learning and machine learning relationships, big data knowledge, AI decision-making processes, knowledge of AI tools, interpretation of AI technologies, critical evaluation, data privacy importance, machine learning knowledge, and evaluation of AI applications in their discipline.

**Data availability statement:** The data supporting the findings of this study are available in Figshare at https://doi.org/10.6084/m9.figshare.30946184.

**Funding:** The author(s) received no specific funding for this work.

**Competing interests:** The authors have declared that no competing interests exist.

## Introduction

Artificial intelligence (AI) has emerged as a pervasive force across multiple fields and is becoming a pivotal tool projected to enhance its functionality through increased understanding and utilization. Although recently, corroborative feedback has begun to surface, particularly in education, where teachers, candidate teachers, and students are key variables. The limited research on relationships between teacher candidates' acceptance of Generative Artificial Intelligence (GenAI), AI literacy (AIL), and demographic variables drives this study. Research shows AIL and self-efficacy significantly influence teacher candidates' acceptance of GenAI tools [1,2]. Positive perceptions of AIL and GenAI correlate with increased willingness to use these tools educationally [3]. Gender, discipline, usage frequency [4], and self-efficacy [5,6] are influential factors. GenAI implementation enhances teacher candidates' digital literacy and problem-solving skills based on usage [7]. Studies show teacher candidates across disciplines have varying perceptions of GenAI, necessitating discipline-specific AIL training [1,2,7]. This study aims to examine the relationship between teacher candidates' GenAI acceptance and AIL levels across disciplines, while offering recommendations. The objectives are to:

- Describe the acceptance levels of teacher candidates from different disciplines regarding GenAI and AIL,

- Identify any existing relationships between these variables,

- Differentiate these variables according to various demographic and/or experiential factors.

- What factors influence the acceptance levels of GenAI,

- What factors influence AIL literacy levels.

### Artificial Intelligence and Its Role in Education

AI has instigated a profound transformation within the educational sector, fundamentally altering teaching, learning, and administrative processes in recent years [5,8–11]. AI-driven systems enhance student performance and alleviate teachers' workloads through applications such as personalized learning paths, intelligent assessment tools, and automated grading [8,12,13]. A critical factor in ensuring the effectiveness and inclusivity of AI is its capacity to deliver content and provide real-time, individualized feedback [14]. Literature underscores AI's role as a supportive tool in pedagogical and creative contexts [15]. Recent investigations into GenAI awareness highlight the need for structured AI literacy development in educational settings. Semerci Şahin et al. [16] devised a Generative Artificial Intelligence Awareness Scale for secondary school students, emphasizing awareness as a construct that includes knowledge, perceived risks, and responsible use of GenAI. While their research focuses on secondary education, the systematic assessment of GenAI awareness is particularly relevant to teacher education, given the limited formal AI training reported by pre-service teachers in the current study.

Despite the ethical challenges associated with the widespread adoption of AI in education [17–19], it remains central to a multidimensional transformation process and plays a pivotal role in shaping future educational models [10,11,18].

AI significantly accelerates and enhances the operations of numerous disciplines through its adaptability to learning environments, which can be customized and optimized for universal effectiveness. It is receptive to systematic feedback and demonstrates rapid and efficient capabilities in both individual and process evaluations [1,20,21]. GenAI models exert both direct and indirect influences on instructional design, learning assessment, and academic writing, thereby introducing novel pedagogical and ethical concepts in teacher training [22,23]. Conversely, UNESCO [24] emphasizes the necessity for the responsible, inclusive, and transparent application of AI in teacher education, highlighting concerns regarding the exacerbation of digital inequalities. In this context, AI should not be perceived merely as a technological tool; its effective integration into teacher education necessitates addressing it alongside various structural components, including competence, ethics, and pedagogy.

Teacher candidates' acceptance of AI is a critical determinant in the widespread adoption and effective implementation of GenAI-based applications within educational settings. Research [25–27] indicates that the most significant predictors of teacher candidates' intentions to utilize GenAI-based educational applications are perceived ease of use and perceived usefulness, which are essential for understanding candidates' behavior in employing AI in the classroom. Furthermore, the acceptance of GenAI is influenced by individual factors such as AIL and self-efficacy [25,26,28]. Acceptance status and concerns regarding AI may also be affected by the user's gender, with Zhang et al. [27] reporting that AI anxiety is more prevalent among female teacher candidates compared to their male counterparts. Additionally, irrespective of gender, all teacher candidates acknowledge that AI can offer personalized content in education, effectively assess student levels, and be tailored to individual needs, while also expressing concerns about security and ethics [29]. In summary, acceptance among teacher candidates is multidimensional, shaped by technological, pedagogical, and psychosocial interactions [25–27,29].

In this study, AI literacy is characterized as the capacity of individuals to comprehend, assess, and effectively engage with AI systems, encompassing an awareness of their foundational principles, limitations, and ethical considerations. This definition is rooted in the framework proposed by Long and Magerko [30], which conceptualizes AI literacy as a set of competencies that include knowledge, skills, and attitudes pertinent to AI. Furthermore, the present study is in alignment with UNESCO's [24] perspective on AI literacy, which underscores the importance of responsible, ethical, and human-centered interaction with AI technologies.

Within the realm of teacher candidate education, AIL is instrumental in enhancing learning outcomes, ensuring transparency and fairness in assessment processes, and processing data in alignment with privacy principles [31,32]. Consequently, AIL should be regarded not as an optional competency in teacher education but as an essential professional competence for educators in the contemporary era.

A correlation is anticipated between the acceptance of technology and the literacy or capability to utilize that technology. Individuals exhibiting high levels of AI acceptance tend to engage with AI more frequently than others and can enhance their literacy over time through learning [33,34]. However, the limited familiarity with GenAI tools and the challenges associated with verifying AI data suggest that acceptance may not always be widespread, conscious, or critical [22,35]. Therefore, it is imperative to examine AI acceptance and literacy concurrently among teacher candidates to elucidate the relationship between their openness to technology and their capacity to employ AI in a pedagogically, ethically, and effective manner.

## Method

### Research model

This study seeks to elucidate pre-service teachers' acceptance of GenAI applications and their literacy levels across various variables, while also facilitating comparisons among them. The research employs explanatory sequential design, a

method within mixed research methodologies. According to Creswell and Plano Clark [36], explanatory sequential design studies integrate quantitative data with qualitative insights to uncover deeper meanings. In this approach, quantitative data are utilized to assess trends and relationships, and qualitative data are employed to elucidate the underlying results of these trends. The objective of this study is to identify the relationships between the dependent variables, namely GenAI acceptance and AIL, and the independent variables, which include gender, grade, department, daily internet usage, AI usage, level of AI proficiency, and the status of receiving AI training, thereby revealing the factors influencing these dependent variables.

## Participants

In this study, written informed consent was obtained from all participants. Quantitative data were collected from 723 pre-service teachers enrolled in various disciplines within the faculty of education at a state university located in the Mediterranean Region.

The study selected pre-service teachers as the focal population due to their formative stage in developing beliefs, attitudes, and intentions regarding educational technologies, including GenAI. During initial teacher education, individuals construct their pedagogical orientations and professional identities, making this group suitable for examining emerging perceptions and acceptance patterns. In contrast, in-service teachers' technology-related beliefs are often influenced by established routines and institutional constraints. Focusing on pre-service teachers enabled the study to capture early-stage perceptions of GenAI before exposure to contextual constraints. This approach aligns with research investigating technology acceptance during teacher preparation programs.

The data collection process was conducted in two phases: qualitative data were gathered between June 24 and July 1, 2025, followed by quantitative data collection between July 2 and July 7, 2025. The descriptive characteristics of the study group are presented in Table 1.

The participant cohort comprises 64.5% female and 35.5% male teacher candidates. In terms of academic progression, 23.7% are first-year students, 31.5% are in their second year, 19.8% are in their third year, and 25% are in their fourth year. The distribution across academic disciplines is as follows: 23% in Special Education (SE), 22.5% in Elementary Mathematics Education (EME), 17.8% in Turkish Language Teaching (TLT), 13.4% in Social Studies Education (SSE), 13.3% in Elementary Science Education (ESE), and 10% in English Language Teaching (ELT). Regarding daily internet usage, 54.4% of participants use the internet "occasionally," 25.3% "rarely," 18.6% "frequently," and only 1.7% "never." Concerning daily engagement with AI applications, 54.6% use them for 4–6 hours, 32.9% use them for 7 hours or more, 11.1% for 1–3 hours, and a mere 1.4% for less than 1 hour. These data suggest that a significant proportion of teacher candidates from various disciplines engage extensively with GenAI.

A total of 65.4% reported utilizing a single GenAI tool, while 24.1% employed two tools, 7.3% used three, and 1.9% utilized four tools. A minority, 1.2%, did not engage with any GenAI applications. ChatGPT emerged as the predominant GenAI tool, with a usage rate of 93.5% among the group. Additionally, 24.1% of participants reported using Copilot, 12.7% used Gemini, 8.2% employed Grammarly, and 3.7% utilized DeepSeek. A small percentage engaged with other applications, including Grok, Midjourney, DALL·E, Quillbot, Character, Canva, Leonardo, and Duolingo. These statistics underscore ChatGPT as the most frequently used AI application among teacher candidates across various disciplines. Regarding GenAI proficiency,

48.1% of the study group identified as having "intermediate" proficiency, 40.4% as "good," 5.9% as "beginner," and 5.5% as "expert." This distribution indicates that the majority of the study group perceives themselves as possessing good or intermediate proficiency in GenAI. However, this self-assessed proficiency contrasts with the status of GenAI training, as only 6.1% of participants have received prior GenAI training, while 93.9% have not undergone any formal GenAI training.

The qualitative data for the study was collected from a total of 48 pre-service teachers, with 8 volunteers from each discipline, selected from the group where quantitative data were gathered. For each discipline, interviews were conducted

**Table 1. Participant Characteristics.**

| Variables | Status | f | % |
|---|---|---|---|
| Gender | Female | 466 | 64.5 |
| | Male | 257 | 35.5 |
| Class | 1 | 171 | 23.7 |
| | 2 | 228 | 31.5 |
| | 3 | 143 | 19.8 |
| | 4 | 181 | 25 |
| Disciplines | B.A. in English Language Teaching (ELT) | 72 | 10 |
| | B.S. in Special Education (SE) | 166 | 23 |
| | B.S. in Social Studies Education (SSE) | 97 | 13.4 |
| | B.S. in Elementary Mathematics Education (EME) | 163 | 22.5 |
| | B.S.in Elementary Science Education (ESE) | 96 | 13.3 |
| | B.S. in Turkish Language Teaching (TLT) | 129 | 17.8 |
| Daily Internet Usage | Never | 12 | 1.7 |
| | Rarely | 183 | 25.3 |
| | Occasionally | 393 | 54.4 |
| | Frequently | 135 | 18.6 |
| Daily GenAI Usage (hours/day) | Less than 1 hour | 10 | 1.4 |
| | 1-3 hours | 80 | 11.1 |
| | 4-6 hours | 395 | 54.6 |
| | 7 hours and more | 238 | 32.9 |
| Count of GenAI Tools Utilized | 0 | 9 | 1.2 |
| | 1 | 473 | 65.4 |
| | 2 | 174 | 24.1 |
| | 3 | 53 | 7.3 |
| | 4+ | 14 | 1.9 |
| GenAI Tools Employed | ChatGPT | 676 | 93.5 |
| | Copilot | 174 | 24.1 |
| | Gemini | 92 | 12.7 |
| | Grammarly | 59 | 8.2 |
| | DeepSeek | 27 | 3.7 |
| | Grok | 21 | 2.9 |
| | Midjourney | 18 | 2.5 |
| | DALL·E | 8 | 1.1 |
| | Quillbot | 4 | 0.5 |
| | Character | 2 | 0.3 |
| | Canva | 2 | 0.3 |
| | Others (Leonardo, Duolingo, Gamma, Napkin, Sider, Seart, Bing, PopAi) | 8 | 1.1 |
| Proficiency in GenAI Usage | Beginner | 43 | 5.9 |
| | Intermediate | 348 | 48.1 |
| | Good | 292 | 40.4 |
| | Expert | 40 | 5.5 |
| Participating in GenAI Training | Yes | 44 | 6.1 |
| | No | 679 | 93.9 |

with 8 pre-service teachers in total— 2 from each grade level, consisting of one female and one male. The interviews were conducted by the same researcher and each interview lasted between 40 and 65 minutes. While the qualitative sample (n = 48) is smaller than the quantitative sample, this size is suitable for qualitative research, which emphasizes depth and richness over numerical representation [37]. Thematic saturation was achieved when no new codes emerged from subsequent interviews, and additional data ceased to provide insights to the analytical framework [38,39]. Representativeness was addressed through purposive sampling to ensure analytical rather than statistical representativeness. Participants were selected to capture variation in characteristics pertinent to the research questions, enhancing the credibility and transferability of findings [40,41]. This approach aligns with qualitative research principles that emphasize diverse perspectives for robust interpretation.

## Ethical considerations

This study was conducted in accordance with the ethical principles governing research involving human participants in educational settings. After obtaining permission from the developers of the measurement instruments, the overall study protocol was reviewed and approved by the Akdeniz University Social and Human Sciences Scientific Research and Publication Ethics Committee (decision no. 410, dated 23 June 2025). Additional authorization was obtained from the relevant faculty of education prior to data collection. Participation was voluntary, and informed consent was obtained from all participants before the study was conducted.

## Data collection tools

The quantitative data of the research were collected using three measurement tools: The Information Form (gender, grade level, daily internet usage duration, etc.) prepared by the researchers and two different scales.

**Information Form:** This form was developed by the researchers to ascertain the characteristics of the study group. It comprises questions aimed at identifying the gender, class, department, daily internet usage, frequency of AI usage (hours/day), number of GenAI applications used, specific GenAI applications employed, AI usage proficiency level, and AI training received by the teacher candidates. GenAI

**Acceptance Scale:** The scale, developed by Karaoğlan Yılmaz et al. [42], was employed to evaluate the GenAI acceptance levels within the study group. It is a 5-point Likert scale consisting of 20 items in its most recent version. The scale encompasses four subscales: Performance Expectancy, Effort Expectancy, Facilitating Conditions, and Social Influence. The scale's reliability is indicated by a Cronbach's alpha coefficient of 0.95. Scores on the GenAI Acceptance Scale range from a minimum of 20 to a maximum of 100.

**AIL Scale:** This scale, developed by Laupichler et al. [43] and adapted into Turkish by Karaoğlan Yılmaz and Yılmaz [44], comprises 31 items on a 7-point Likert scale. It includes three subscales: Technical Understanding, Critical Evaluation, and Practical Application. The scale's reliability is demonstrated by a Cronbach's alpha coefficient of 0.957. The score range is 31-217, with higher scores signifying greater AIL among students.

Permission was secured from the researchers who developed the scales for the deployment of data collection instruments. Subsequently, the comprehensive study plan was submitted to the ethics committee of the institution where the researchers were affiliated. Upon receiving approval, authorization was sought from the relevant faculty of education to administer the study to students. Following approval, the study was conducted in person on a voluntary basis.

The results indicated that participants' levels of acceptance of GenAI and AIL varied according to their departmental affiliation. To explore the potential reasons for these departmental differences in greater depth, qualitative interview instruments were developed. The qualitative data for this research was gathered using interview forms developed by the researchers. Initially, the interview form comprised ten questions. These ten open-ended questions, pertaining to the acceptance of GenAI and AIL, were reviewed by three experts in the field of computer and technology, four experts in

assessment and evaluation, and two language experts. Following this review, three questions were eliminated due to a lack of consensus among the experts, and two questions were merged into one, resulting in a final version consisting of six questions.

## Data analysis

Of the 747 forms collected from participants, 24 were excluded from evaluation due to issues such as unreadability, blank items, or random completion, resulting in 723 forms deemed suitable for analysis. The data analysis was conducted using the SPSS 25.0 software. To assess whether the data adhered to a normal distribution, which is essential for describing dependent and independent variables, the skewness and kurtosis values of the variables were examined to ensure they fell within the ±2 range [45] and the histogram was checked for symmetrical distribution [46]. The Shapiro-Wilk and Kolmogorov-Smirnov tests, typically employed in normality distribution assessments, were not utilized due to their tendency to identify even minor deviations as statistically significant, particularly in large samples [45,46].

Examining the normality distributions of the dependent and independent variables in this study, it was found that all variables, except for the GenAI training variable, conformed to the normality value range specified by Tabachnick and Fidell [45] (Table 2). Consequently, parametric tests, including the independent samples t-test, ANOVA, and Pearson correlation, were employed for variables such as gender, class, department, daily internet usage, and the frequency and number of GenAI usage, as appropriate for the number of variables. For the GenAI training variable, the Mann-Whitney U test, a nonparametric equivalent of the independent samples t-test, was applied.

In the ANOVA test, the Levene's test statistic was evaluated to ensure the validity of the analysis. For $p < .05$, the significance value between groups was determined; for $p > .05$, the significance was assessed using the result of Welch's F test. In instances where a significant difference was identified with the ANOVA test, homogeneity was examined to ascertain between which units the difference existed. Due to the non-homogeneous distribution of the data, the Dunnett C test was employed to determine the differences between the units.

Furthermore, Eta squared ($\eta^2$) values were assessed to ascertain the effect size of the ANOVA test results, specifically the proportion of total variance explained by the independent variable [47]. The Eta squared value, which ranges from 0 to 1, is interpreted as indicating a small effect if it is 0.01 or less, a medium effect if it is up to 0.06, and a large effect if it is 0.14 or more [48].

In this study, MAXQDA 2024 software was employed for the analysis of qualitative data. The MAXQDA program aids in the identification and categorization of various themes and sub-themes derived from qualitative data [49]. Specifically, it was utilized to elucidate the themes and sub-themes associated with the factors influencing GenAI acceptance levels and AIL.

Following the transcription of the interviews, the researchers independently developed codes and themes aligned with the study's objectives through content analysis, utilizing MAXQDA based on the transcripts. The researchers then engaged in discussions with experts to evaluate the consistency of these codes and themes, who conducted analyses related to them. Miles and Huberman [50] quantify this consistency—termed internal consistency and conceptualized as coder agreement—using the formula: $\Delta = \mathbb{C} \div (\mathbb{C} + \partial) \times 100$. In this formula, $\Delta$ denotes the reliability coefficient, $\mathbb{C}$ represents the number of items/terms agreed upon, and $\partial$ signifies the number of items/terms not agreed upon. The coding audit, which assesses internal consistency, revealed a 96% agreement among coders. After necessary adjustments, the finalized version of the codes and themes, as referenced in the findings section, was presented.

**Table 2. Flatness and Skewness Status of Variables.**

| Variables | N | Skewness | Kurtosis |
|---|---|---|---|
| GenAI Acceptance | 723 | −0.328139 | 0.271675 |
| AIL | 723 | 0.211029 | −0.376364 |

## Findings

### Findings related to GenAI Acceptance

Analysis of the mean score derived from the GenAI Acceptance Scale reveals an average of 76.33, with a standard deviation of 12.296. These results indicate that the participants exhibit an acceptance level that surpasses the average (Fig 1).

The findings from the independent samples t-test, conducted to assess whether GenAI acceptance status differs by gender (Table 3), reveal no statistically significant difference (t (721) = -0.260, p > 0.05). This outcome suggests that GenAI acceptance status does not differ between genders, indicating that both female and male teacher candidates exhibit comparable levels of acceptance.

A one-way analysis of variance (ANOVA) was conducted to assess the acceptance status of GenAI within the context of the departments in which the participants received their education (Table 4).

The analysis results indicate that the level of GenAI acceptance varies across different departments. Although the effect size is low to moderate, the level of statistical significance achieved suggests that there is a statistically reliable difference in GAI acceptance across categories. ($F_{(5-716)}$ = 4.029, p<0.01, η2: 0,03). The Dunnett C test, conducted to identify which departments exhibited differences, revealed that the average score of students in the ESE department (=80.24) was significantly higher than the average scores of students in the SE ($\overline{X}$=75.04), EME ($\overline{X}$=74.96), and TLT ($\overline{X}$=74.82) departments.

In other words, ESE students, with the highest average scores, demonstrate a higher level of GenAI acceptance compared to students in the Special Education and Primary Mathematics departments, as well as those in the TLT department, who have the lowest average scores.

When examining GenAI acceptance status in relation to class variables, differences were observed in participants' GenAI acceptance status with a small effect size ($F_{(3-719)}$ = 4.806, p<0.01, η2: 0,02). The Dunnett C test results indicate that the grade point average of 4th-grade students ($\overline{X}$=79.09) is significantly higher than that of both 1st-grade ($\overline{X}$=75.22) and 2nd-grade students ($\overline{X}$=74.78). Additionally, the 4th grade exhibited the highest GenAI acceptance level, while the 2nd grade exhibited the lowest.

When comparing daily internet usage time with GenAI acceptance level, no significant difference was found ($F_{(3-719)}$ = 2.785, p>0.01). In other words, participants' daily internet usage time does not influence their GenAI acceptance status. Although there is no significant difference, the average score of participants who frequently use the internet ($\overline{X}$=78.37) is the highest, while the average score of participants who never use the internet ($\overline{X}$=70.92) is the lowest.

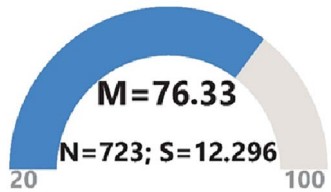

**Fig 1. Participants' level of GenAI acceptance.**

**Table 3. Results of t-test comparing participants' GenAI acceptance status according to gender variable.**

| Gender | N | $\overline{X}$ | S | t | p |
|---|---|---|---|---|---|
| Female | 466 | 76.24 | 12.048 | −.260 | .795 |
| Male | 257 | 76.49 | 12.755 | | |

**Table 4. Results of the One-Way Analysis of Variance (ANOVA) Comparing Participants' Department, Class, Internet Usage, GenAI Usage Duration, Number of GenAI Applications, and GenAI Usage Competence Level with GenAI Acceptance Level.**

| Variable | Status | N | $\overline{X}$ | S | F | p | Difference (Dunnett C) | η² |
|---|---|---|---|---|---|---|---|---|
| Disciplines | ELT | 72 | 76.40 | 10.272 | 4.029 | .003 | 5−2 5−4 5-6 | 0.03 |
| | SE | 166 | 75.04 | 13.393 | | | | |
| | SSE | 97 | 78.90 | 13.407 | | | | |
| | EME | 163 | 74.96 | 9.529 | | | | |
| | ESE | 96 | 80.24 | 12.956 | | | | |
| | TLT | 129 | 74.82 | 12.862 | | | | |
| | Total | 723 | 76.33 | 12.296 | | | | |
| Class | 1 | 171 | 75.22 | 12.468 | 4.806 | .003 | 1-4 2-4 | 0.02 |
| | 2 | 228 | 74.78 | 12.775 | | | | |
| | 3 | 143 | 76.62 | 11.891 | | | | |
| | 4 | 181 | 79.09 | 11.415 | | | | |
| | Total | 723 | 76.33 | 12.296 | | | | |
| Daily Internet Usage | Never | 12 | 70.92 | 17.149 | 2.785 | 0.4 | | |
| | Rarely | 183 | 74.97 | 11.969 | | | | |
| | Occasionally | 393 | 76.42 | 11.946 | | | | |
| | Frequently | 135 | 78.37 | 13.003 | | | | |
| | Total | 723 | 76.33 | 12.296 | | | | |
| Daily GenAI Usage | Less than 1 hour | 10 | 61.70 | 17.455 | 20.341 | .000 | 1-4 2-3 2-4 | 0.08 |
| | 1-3 hours | 80 | 69.10 | 11.047 | | | | |
| | 4-6 hours | 395 | 76.32 | 11.957 | | | | |
| | 7 hours and more | 238 | 79.38 | 11.602 | | | | |
| | Total | 723 | 76.33 | 12.296 | | | | |
| Count of GenAI Tools Utilized | 0 | 9 | 62.44 | 16.508 | 9.057 | .000 | 1.2.3.4-5 | 0.05 |
| | 1 | 473 | 75.45 | 12.399 | | | | |
| | 2 | 174 | 77.59 | 11.526 | | | | |
| | 3 | 53 | 78.74 | 10.145 | | | | |
| | 4+ | 14 | 89.86 | 7.999 | | | | |
| | Total | 723 | 76.33 | 12.296 | | | | |
| Proficiency in GenAI Usage | Beginner | 43 | 70.60 | 15.131 | 15.756 | .000 | 1-3.4 2-3.4 | 0.06 |
| | Intermediate | 348 | 74.11 | 11.479 | | | | |
| | Good | 292 | 78.93 | 11.752 | | | | |
| | Expert | 40 | 82.73 | 13.295 | | | | |
| | Total | 723 | 76.33 | 12.296 | | | | |

In analyzing the correlation between daily AI usage duration and the level of acceptance of GenAI, it was observed that participants' acceptance levels varied significantly with different AI usage time representing a medium effect size ($F_{(3-719)}$ = 20.341, $p < 0.01$, η2: 0,08). The results of the Dunnett C test indicate that the mean acceptance score of participants who engaged with GenAI for 7 hours or more ($\overline{X}$=79.38) was significantly higher than that of participants who used GenAI for less than 1 hour ($\overline{X}$=61.7) or between 1 and 3 hours ($\overline{X}$=69.1). Similarly, the mean score of participants using GenAI for 4–6 hours ($\overline{X}$=76.32) was significantly greater than that of those using it for 1–3 hours.

Furthermore, when examining the relationship between the number of different GenAI applications utilized and GenAI acceptance, a significant association was identified ($F_{(4-718)}$ = 9.057, $p < 0.001$, η2: 0,05). Although the effect size was

small-to-medium, the results indicate that GAI acceptance levels differ significantly based on the variety of AI applications used. Specifically, the number of GenAI applications employed by participants influences their acceptance levels. The Dunnett C test results reveal that the acceptance level of participants using four different GenAI applications ($\overline{X}$=89.86) is significantly higher than the average scores of those who do not use GenAI applications ($\overline{X}$=62.44) and those using one, two, or three applications (respectively $\overline{X}$=75.45, $\overline{X}$=77.59, $\overline{X}$=78.74).

In comparing GenAI acceptance levels with AI usage proficiency, a significant difference was detected with a medium effect size ($F_{(3-719)}$ = 15.756, $p < 0.001$, η2: 0,06). According to the Dunnett C test results, the average scores of participants who self-assessed as good (=78.93) and expert ($\overline{X}$=82.73) were significantly higher than those of participants at the beginner ($\overline{X}$=70.6) and intermediate ($\overline{X}$=74.11) levels. Notably, the average score increases with higher levels of competence.

A Mann-Whitney U test was conducted to assess whether participants' receipt of training on GenAI usage influenced their acceptance levels (Table 5).

Accordingly, the relationship between the two variables is not statistically significant (U = 12764.000; $p > 0.05$). Therefore, whether or not individuals have received training related to GenAI does not result in any difference in GenAI acceptance.

## Findings related to AIL

The average score obtained from the AIL scale is 113.64, with a standard deviation of 35.134. This average score suggests that the participants possess an AIL level below the mean (Fig. 2).

The independent samples t-test conducted to assess whether the AIL level varied by gender revealed a statistically significant difference between the variables (t(721)= −4.301, $p < 0.05$) (Table 6).

The findings indicate that AIL levels vary by gender, with male teacher candidates exhibiting higher average scores and consequently higher literacy levels compared to their female counterparts.

A one-way analysis of variance (ANOVA) was performed to assess whether participants' AIL levels differed based on specific variables (Table 7).

Evaluating participants' AIL levels through ANOVA testing within the context of their respective educational departments, it was observed that AIL levels significantly varied across departments. The mean AIL level score differed according to the participant's department with a small effect size ($F_{(5-716)}$ = 3.789, $p < 0.05$, η2: 0.03). The Dunnett C test indicated that the mean score of students in the ELT department ($\overline{X}$=103.63) was significantly lower than those of students in the

**Table 5. Mann-Whitney U Results Comparing Participants' Acceptance of GenAI Based on Receipt of GenAI Training.**

| GenAI Training | N | Mean Rank | Rank Total | U | p |
|---|---|---|---|---|---|
| Yes | 44 | 411.41 | 18102.00 | 12764.000 | .105 |
| No | 679 | 358.80 | 243624.00 | | |

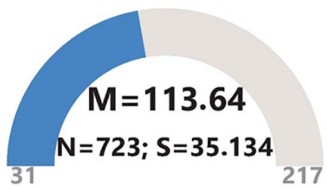

**Fig 2. Participants' AIL level.**

**Table 6. t-Test Results Comparing Participants' AI Acceptance Literacy Levels According to Gender Variable.**

| Gender | N | X | S | t | p |
|--------|-----|--------|--------|--------|------|
| Female | 466 | 109.52 | 34.181 | −4.301 | .000 |
| Male | 257 | 121.12 | 35.667 | | |

**Table 7. Results of One-Way Analysis of Variance (ANOVA) Concerning the Comparison of Participants' Department, Class, Internet Usage, GenAI Usage Duration, Number of GenAI Applications, and GenAI Usage Proficiency Level with AIL Level.**

| Variable | Status | N | $\overline{X}$ | S | F | p | Difference (Dunnett C) | η2 |
|----------|--------|-----|--------|--------|--------|------|------------------------|------|
| Disciplines | ELT | 72 | 103.63 | 31.127 | 3.786 | .002 | 1-3.5 | 0.03 |
| | SE | 166 | 110.58 | 36.149 | | | 2-5 | |
| | SSE | 97 | 118.95 | 36.979 | | | | |
| | EME | 163 | 112.10 | 31.106 | | | | |
| | ESE | 96 | 124.34 | 37.294 | | | | |
| | TLT | 129 | 113.16 | 35.771 | | | | |
| | Total | 723 | 113.64 | 35.134 | | | | |
| Class | 1 | 171 | 110.06 | 34.416 | 3.457 | .016 | 2-4 | 0.01 |
| | 2 | 228 | 109.98 | 33.178 | | | | |
| | 3 | 143 | 116.19 | 34.929 | | | | |
| | 4 | 181 | 119.63 | 37.572 | | | | |
| | Total | 723 | 113.64 | 35.134 | | | | |
| Daily Internet Usage | Never | 12 | 105.83 | 28.077 | .411 | .745 | – | |
| | Rarely | 183 | 114.38 | 36.568 | | | | |
| | Occasionally | 393 | 112.89 | 33.145 | | | | |
| | Frequently | 135 | 115.51 | 39.306 | | | | |
| | Never | 723 | 113.64 | 35.134 | | | | |
| Daily GenAI Usage | Less than 1 hour | 10 | 109.20 | 41.317 | .799 | .495 | – | |
| | 1-3 hours | 80 | 112.41 | 28.457 | | | | |
| | 4-6 hours | 395 | 112.29 | 35.883 | | | | |
| | 7 hours and more | 238 | 116.48 | 35.675 | | | | |
| | Total | 723 | 113.64 | 35.134 | | | | |
| Count of GenAI Tools Utilized | 0 | 9 | 111.67 | 38.118 | 3.252 | .012 | 2-5 | 0.02 |
| | 1 | 473 | 111.19 | 34.919 | | | | |
| | 2 | 174 | 115.72 | 35.418 | | | | |
| | 3 | 53 | 122.72 | 34.344 | | | | |
| | 4+ | 14 | 137.36 | 27.587 | | | | |
| | Total | 723 | 113.64 | 35.134 | | | | |
| Proficiency in GenAI Usage | Beginner | 43 | 98.07 | 33.922 | 14.142 | .000 | 1-3.4 | 0.06 |
| | Intermediate | 348 | 107.79 | 31.653 | | | 2-3.4 | |
| | Good | 292 | 120.32 | 36.448 | | | | |
| | Expert | 40 | 132.55 | 38.904 | | | | |
| | Total | 723 | 113.64 | 35.134 | | | | |

SSE ($\overline{X}$=118.95) and ESE ($\overline{X}$=124.34) departments. Similarly, the average score of SE students ($\overline{X}$=110.58) was significantly lower than that of ESE students ($\overline{X}$=124.34). Furthermore, ESE students ($\overline{X}$=124.34) exhibited the highest AIL level, whereas ELT students ($\overline{X}$=103.63) demonstrated the lowest.

The AIL level also varied according to the class variable ($F_{(3-719)}$ = 3.457, p<0.05, η2 =.01), despite a small effect size. According to the Dunnett C test results, 4th-grade students, with the highest literacy score average ($\overline{X}$=119.63), had a significantly higher AIL level than 2nd-grade students, who had the lowest score average ($\overline{X}$=109.98).

In contrast, when comparing AIL levels with daily internet usage time, no significant difference was observed ($F_{(3-719)}$ = 0.411, p>0.001), indicating that participants' daily internet usage time does not affect their AIL status.

Similarly, GenAI usage time did not alter the AIL level ($F_{(3-719)}$ =.799, p>0.05). However, the average score of participants who used GenAI for 7 hours or more ($\overline{X}$=116.48) was higher than that of participants who used it for less than one hour ($\overline{X}$=109.2).

The AIL level demonstrated a significant difference based on the number of different GenAI applications utilized, despite a small effect size ($F_{(4-718)}$ = 9.057, p<0.05, η2 =.02). The average score of participants who used four different GenAI applications ($\overline{X}$=137.36) was significantly higher than that of participants who used one GenAI application ($\overline{X}$=111.19), suggesting that participants who used four different GenAI applications possessed higher AIL than those who used one.

The AIL level also varied according to the GenAI proficiency level has a moderate effect size ($F_{(3-719)}$ = 14.142, p<0.05, η2: 0,06). According to the Dunnett C test results, the mean scores of participants who classified themselves as beginner ($\overline{X}$=98.07) and intermediate ($\overline{X}$=107.79) were significantly lower than those of participants who classified themselves as good ($\overline{X}$=120.32) and expert ($\overline{X}$=132.55). Additionally, the average score increased with the level of competence.

The Mann-Whitney U test was conducted to determine whether participants' training status regarding GenAI use affected their AIL level, revealing a significant relationship between the two variables (U = 10319.500; p<0.05) (Table 8).

## The relationship between GenAI acceptance and AIL

The Pearson correlation test (Table 9, Fig 3) conducted to examine the relationship between participants' GenAI acceptance status and AIL level reveals a low but significant positive bidirectional relationship between the variables (r =.297, p < 0.01). This indicates that changes in the variables are parallel. In other words, as the AIL level increases, GenAI acceptance also increases, and vice versa.

The themes regarding the factors affecting the acceptance of GenAI by pre-service teachers are presented in Fig 4.

Fig 4, it is seen that the findings are grouped under 9 themes: daily use, problem solving, learning to use the applications, use by mentors, ability to receive help from others, being proficient in usage, productivity, serving discipline-specific

**Table 8. Results of the t-test comparing participants' AIL levels according to the variable of receiving GenAI training.**

| GenAI training | N | Mean Rank | Rank Total | U | p |
|---|---|---|---|---|---|
| Yes | 44 | 466.97 | 20546.50 | 10319.500 | .001 |
| No | 679 | 355.20 | 241179.50 | | |

The mean rank of participants who received training (466.97) is significantly higher than that of those who did not receive training (355.2). This finding suggests that training related to GenAI enhances the level of AIL.

**Table 9. Correlation Between GenAI Acceptance and AIL.**

| | | AIL |
|---|---|---|
| GenAI Acceptance | Pearson r | .297 ** |
| | p | .000 |
| | N | 723 |

**Correlation is significant at the 0.01 level (2-tailed).

 

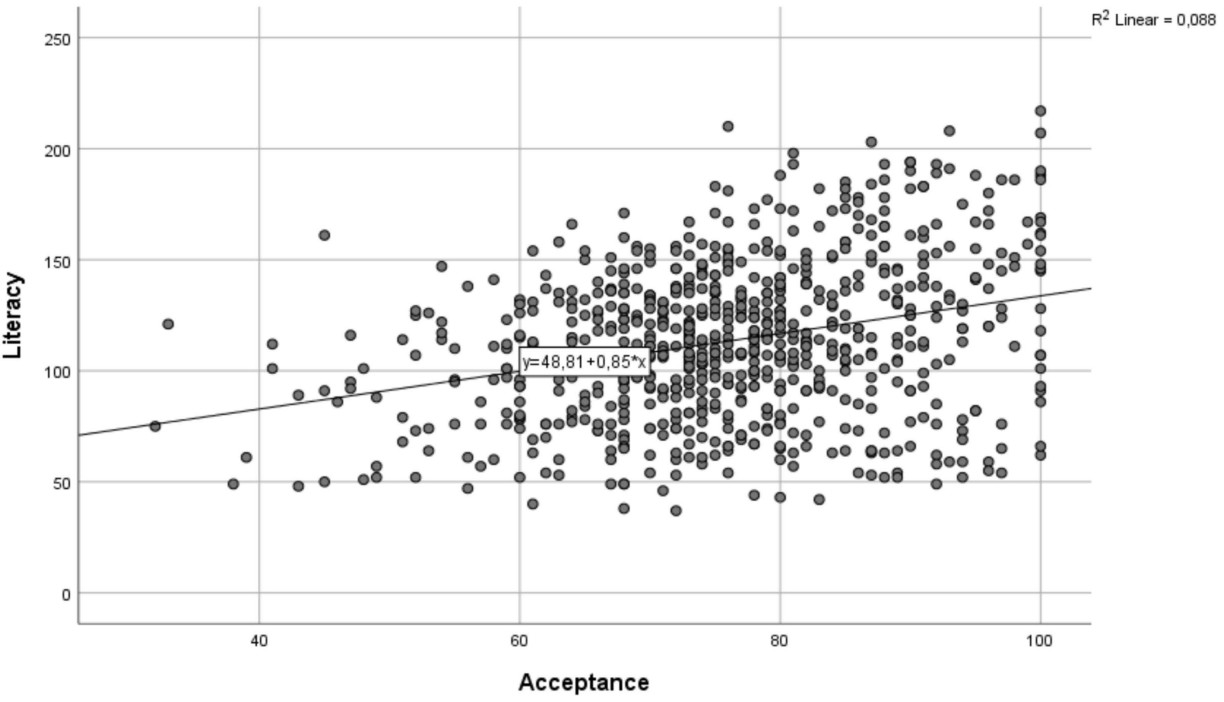

**Fig 3. Correlation between GenAI acceptance and AILFactors Affecting the Acceptance of GenAI.**

skills, and finishing tasks more quickly. While the overarching themes are consistent, the sub-themes among teacher candidates exhibit variation. Specifically, positive statements regarding a particular discipline are grouped under a common theme, whereas candidates from a different discipline have articulated negative perspectives. For instance, candidates from the ESE and EME disciplines reported that their mentors' utilization of AI applications, their support in integrating these applications into lessons, the introduction of AI-related courses, and the provision of assistance, when necessary, had beneficial effects. Conversely, candidates from the SE and TLT disciplines expressed negative views within this theme. Furthermore, it was noted that ESE teacher candidates with high levels of GenAI acceptance, unlike TLT candidates with lower acceptance levels, offered insights into the sub-themes of problem-solving and mentor usage. While the theme of daily use is prevalent across all disciplines, the theme that diverges pertains to serving discipline-specific skills. In this context, ESE and EME candidates indicated that GenAI tools are functional, usable, and capable of enhancement under the serving discipline-specific skills theme. In contrast, candidates from the SE, TLT, ELT, and SSE disciplines contended that GenAI tools are unsuitable for their fields, inadequate in recognizing cultural differences, and not amenable to improvement.

### Factors Affecting AIL

The themes related to the factors influencing the AIL levels of prospective teachers are presented in Fig 5.

While the perspectives of teacher candidates do not converge into common themes regarding AIL, variations are evident in the sub-themes. For instance, candidates from the ESE program with a high level of AIL reported possessing, utilizing, and interpreting technical knowledge of machine learning. In contrast, candidates from the ELT program with a low level of AIL indicated a lack of opinions on the technical aspects of AI. Additionally, it is noted that, unlike their ELT counterparts, ESE candidates articulated their understanding of the relationship between deep learning and machine

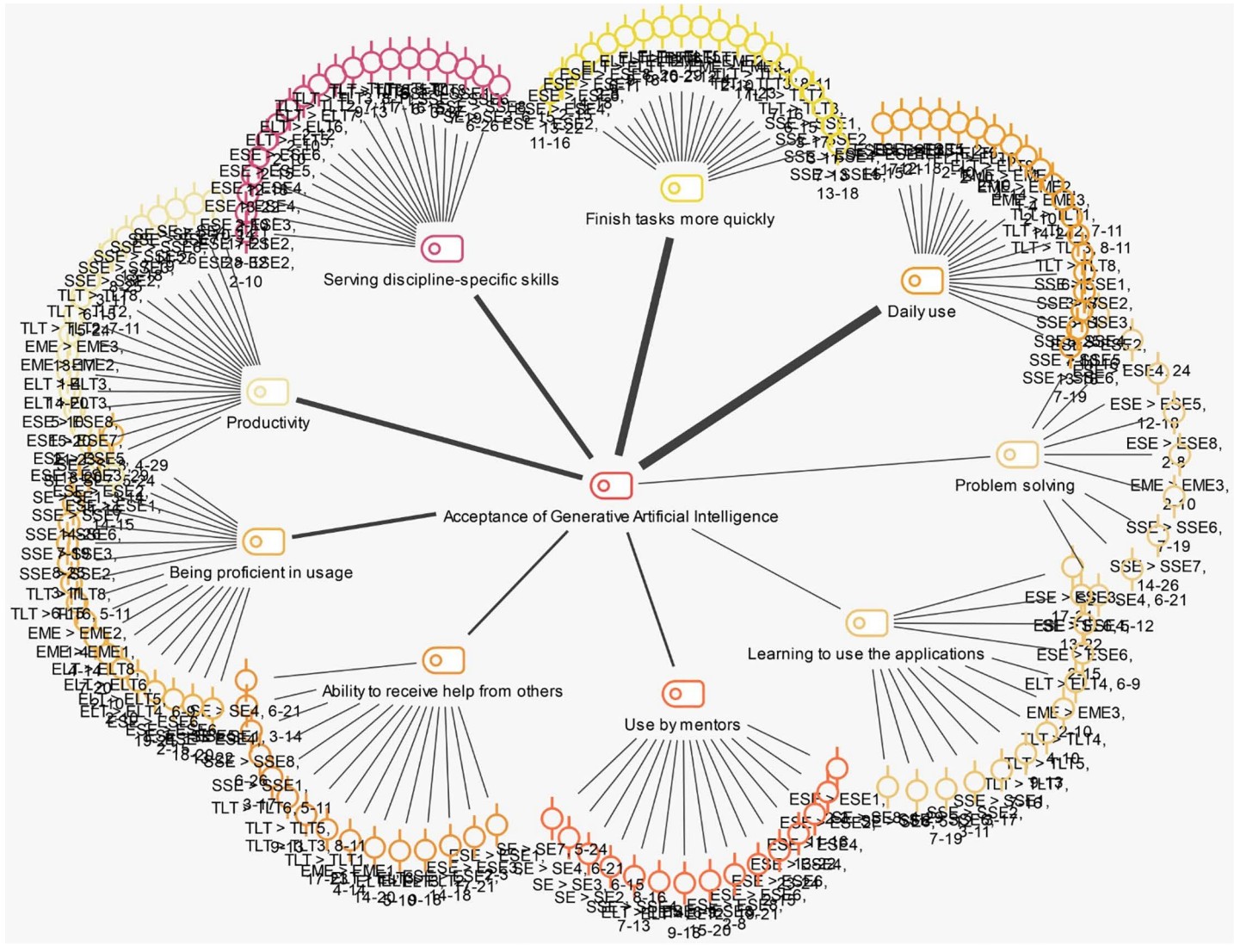

**Fig 4. Factors affecting the acceptance of AI.**

learning, as well as their comprehension of decision-making processes in AI applications. Moreover, a theme common to both disciplines is the utilization of AI support in daily life. The theme where divergent opinions are observed pertains to the technical knowledge of machine learning.

## Results, Discussion and Suggestions

The study revealed that teacher candidates generally demonstrated a positive reception towards GenAI applications, although the average AIL level remained below six. Previous research [51,52] corroborates these findings, indicating that while teacher candidates perceive GenAI as beneficial and practical, their acceptance levels are high, yet AIL predominantly remains at a fundamental level.

It was concluded that prospective teachers from different disciplines have an above-average level of acceptance in using GenAI applications, but that this acceptance varies by discipline. The qualitative findings of the study explain and

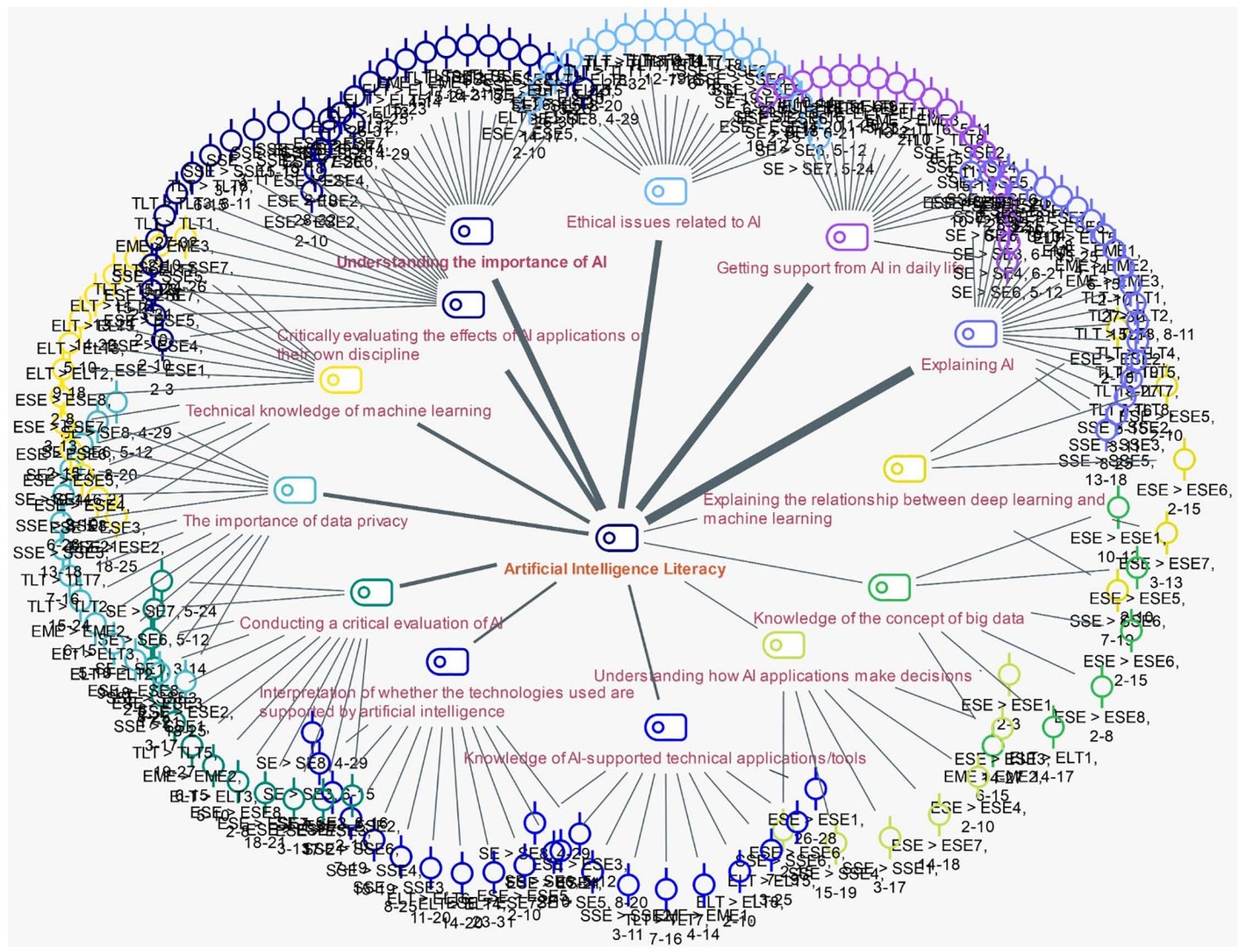

**Fig 5. Factors affecting AILFigure 5; it can be seen that thirteen themes were identified: understanding the importance of AI, ethical issues related to AI, getting AI support in daily life, explaining AI, explaining the relationship between deep learning and machine learning, knowledge of the concept of big data, understanding how AI applications make decisions, knowledge of AI-supported technical applications/tools, interpretation of whether the technologies used are supported by AI, conducting a critical evaluation of AI, the importance of data privacy, technical knowledge of machine learning, and critically evaluating the effects of AI applications on their discipline.**

support the quantitative findings. It was observed that prospective teachers with a high level of GenAI acceptance also expressed positive opinions during interviews, while those with a low level of GenAI acceptance held more negative views. The positive opinions indicated that GenAI tools are functional, usable, and improvable, whereas the negative opinions stated that the GenAI tools are not suitable for their intended purpose, are insufficient in distinguishing cultural differences, and are not capable of being improved. Furthermore, it was determined that teacher candidates from various disciplines exhibited an above-average acceptance of GenAI applications. Both female and male teacher candidates from different disciplines displayed similar attitudes towards the utilization of GenAI applications, with no significant gender differences observed. An examination of the departments in which teacher candidates from diverse disciplines are enrolled

revealed variability in the acceptance of AI applications. ESE teacher candidates demonstrated a higher level of acceptance, whereas those in the SE, EME, and TLT fields exhibited lower acceptance levels.

Similarly, acceptance of GenAI applications varied according to the grade level being taught, with fourth-grade teacher candidates showing a higher level of acceptance compared to other grades. No correlation was identified between the daily internet usage time of teacher candidates and their acceptance of GenAI applications, indicating that acceptance levels are consistent regardless of daily internet usage duration. However, significant differences in AI application acceptance were noted when comparing the time allocated to AI applications. Teacher candidates who dedicated more time to GenAI applications and received support exhibited higher levels of AI acceptance, suggesting a positive correlation between time investment and acceptance. This is attributed to the increase in acceptance as the personal time allocated to AI applications increases. Additionally, the variety of different AI applications utilized by teacher candidates significantly influenced their acceptance of AI applications. It was observed that the greater the diversity and number of GenAI applications used by teacher candidates, the more they demonstrated an attitude of acceptance towards GenAI applications.

Various and frequently utilized GenAI applications significantly enhance teacher candidates' digital literacy and critical thinking skills, with this effect being further reinforced by personal experiences and applications [7,51,53]. The proficiency of candidate teachers in employing AI applications influences their acceptance of AI. Those who identify as experts or proficient exhibit a higher level of acceptance compared to those with beginner or intermediate knowledge. The acceptance of AI applications among candidate teachers does not vary based on whether they have received AI training.

A significant finding of this study is that only 6.1% of participants reported receiving formal training in artificial intelligence. This low percentage suggests pre-service teachers engage with GenAI informally and independently, rather than through systematic support from teacher education programs. Previous research indicates that lack of structured training may lead to superficial understanding, inconsistent skill development, and limited awareness of ethical and pedagogical implications of AI use in education [24,30]. This finding underscores a misalignment between the rapid proliferation of GenAI technologies and teacher education curricula's capacity to prepare future teachers for responsible and effective use.

The findings can be interpreted through established technology adoption frameworks. From the Technology Acceptance Model (TAM) perspective, participants' positive perceptions of GenAI align with key determinants—perceived usefulness and ease of use—which influence users' attitudes and adoption intentions [54,55]. Additionally, patterns in participants' accounts resonate with Diffusion of Innovations Theory: perceived instructional value corresponds with relative advantage, perceived alignment with teaching practices reflects compatibility, and emphasis on experimenting in low-stakes contexts relates to trialability [56]. These theoretical linkages suggest that pre-service teachers' emerging acceptance of GenAI is informed by individual evaluations of utility and usability (TAM) and by innovation attributes facilitating experimentation and adoption decisions [56]. While this study does not aim to formally test TAM or Diffusion of Innovations Theory, these frameworks provide a useful lens for interpreting the observed acceptance patterns [54,56].

The findings of this study align with and expand upon recent research on GenAI awareness and its pedagogical application in education. Semerci Şahin et al. [16] conceptualized GenAI awareness as a multidimensional construct including knowledge, perceived risks, and responsible use, emphasizing systematic AI literacy development. Similarly, Sarıkahya et al. [57] showed that faculty members' experiences with ChatGPT reveal pedagogical opportunities and literacy gaps when formal training is insufficient. Building on these insights, this study offers a complementary perspective by examining pre-service teachers and demonstrating how limited AI training may influence perceptions and intentions regarding GenAI use before classroom practice. This discussion situates the findings within emerging, discipline-sensitive research on GenAI in education. The AIL among pre-service teachers is notably low. The qualitative findings within the AIL dimension of the study elucidate and corroborate the quantitative results pertaining to AIL. While common themes have been identified, the sub-themes exhibit variation, and it has been observed that disciplines with lower AIL levels often hold negative perceptions of AIL. In numerically oriented departments, such as EME and ESE, this is attributed to their experiences with

machine learning, encompassing detailed knowledge and interpretative skills. Conversely, in verbally oriented departments, including SE, TLT, SSE, and ELT, pre-service teachers cite a lack of experience and the perception of irrelevance to their discipline as reasons for their low AIL. Furthermore, the level of AIL appears to vary by gender, with male participants demonstrating significantly higher levels of AIL.

This disparity may be attributed to the digital inequalities and social barriers to technology access highlighted by Blikstein [31]. In terms of gender and field differences, some studies indicate that male teacher candidates score higher in AIL, while candidates in science and technology-focused departments exhibit higher acceptance and literacy levels [1,7,58]. AIL varies according to the department in which teacher candidates are enrolled. Candidates in science education demonstrate higher AIL, particularly in ELT, compared to other teaching disciplines. It is noteworthy that ELT students possess the lowest level of AIL. The higher acceptance and literacy levels among Science Education students can be attributed to their programs' greater openness to technology integration [32]. This suggests that disciplinary differences may be shaped by the degree of technology focus in teacher education programs.

The grade level of teacher candidates also affects their AIL, with fourth-grade students exhibiting higher literacy than those in other grade levels, particularly second-grade students. There is no correlation between the daily internet usage time of teacher candidates and the time they spend using AI and their AIL. In other words, the daily time spent on the internet or AI applications does not influence their AIL. The number of different AI applications utilized by teacher candidates is a significant determinant of AIL. Candidates who use four different AI applications demonstrate higher literacy than those who use two or fewer applications. The proficiency of teacher candidates in using AI applications impacts their AIL levels. Candidates who identify as experts or proficient exhibit higher literacy than those who report beginner or intermediate knowledge. Whether teacher candidates have received AI-related training influences their AIL levels, as having received AI training correlates with higher literacy.

Empirical evidence supports the assertion that significant improvements in literacy and acceptance levels are observed among teacher candidates who receive AI training, underscoring the critical importance of pre-service AI education [2,52,53]. Kasneci et al. [59] highlights the frequent use of Generative AI (GenAI) tools, such as ChatGPT, by teachers, although this usage often lacks a foundation in critical information literacy. This indicates that while teacher candidates are inclined to adopt GenAI tools, their understanding of the ethical, social, and pedagogical implications of these tools is notably limited. In this study, a positive but low-level correlation (r =.297) was identified between GenAI acceptance and AIL, suggesting that acceptance does not necessarily equate to knowledge and competence. This finding aligns with the advanced applications of the Technology Acceptance Model by Venkatesh and Davis [55], which posits that acceptance does not inherently lead to a profound understanding, even as perceived benefits and ease of use increase.

Notably, candidates who utilized GenAI applications more frequently exhibited higher acceptance scores; however, only those who received GenAI training demonstrated a significant enhancement in literacy. This finding corroborates the emphasis on "critical AIL" as proposed by Laupichler et al. [43]. In essence, as teacher candidates' AIL levels rise, so does their acceptance of GenAI applications. This outcome is consistent with studies that reveal a positive and significant, albeit generally low to moderate, relationship between AIL and GenAI acceptance [50,58].

In conclusion, AIL, application experience, and discipline-specific training are pivotal in influencing teacher candidates' acceptance of GenAI, while considerations of gender and branch differences are pertinent to discussions of digital equality and inclusive education.

The findings also underscore the necessity of addressing not only the utilization of existing technological tools but also their pedagogical and ethical dimensions. Indeed, UNESCO's [24] policy framework advocates for teachers to serve not only as users but also as technology critics and ethical guides.

A limitation of this study is its exclusive focus on pre-service teachers, which may restrict the generalizability of the findings. Therefore, future research should include in-service teachers examining how professional experience and classroom realities influence AI-related perceptions and practices.

Despite this limitation, pre-service teachers represent a crucial demographic for examination, as their beliefs and intentions regarding technology use are predominantly formed during initial teacher education and may subsequently influence their future instructional practices [60,61]. Conversely, the perceptions of in-service teachers concerning AI are likely to be more significantly affected by contextual and institutional factors, such as curriculum demands, classroom management responsibilities, accountability pressures, and school-level policies, which may result in distinct patterns of AI acceptance and utilization [62,63]. Consequently, the findings of the present study should be primarily interpreted as indicative of emerging perceptions and intentions toward AI use, rather than established classroom practices shaped by extensive professional experience [64].

Beyond adoption challenges, the findings highlight the need for greater focus on ethical considerations of GenAI in teacher education. Critical issues like algorithmic bias, data privacy, and transparency are crucial in educational settings, where AI systems affect instructional decisions, assessment practices, and student opportunities [24,32]. Pre-service teachers without formal AI training may be ill-equipped to assess biased outputs, protect student data, or understand the opaque decision-making in GenAI systems. This emphasizes the need to integrate ethical reflection, critical AI literacy, and discussions of responsibility within teacher preparation programs, rather than viewing AI merely as a technical tool.

Moreover, previous research examining in-service teachers and university faculty offers a practice-oriented perspective on the adoption of GenAI. Recent studies suggest that the adoption of GenAI by experienced educators is influenced not only by perceived usefulness and ease of use but also by concerns regarding academic integrity, ethical implications, institutional guidelines, and disciplinary norms [65–67]. Furthermore, research based on diffusion of innovations theory indicates that professional experience and organizational context significantly influence GenAI adoption decisions among academics [68].

In contrast, the findings of the present study reflect the emerging perceptions and intentions of pre-service teachers, which are less constrained by institutional and classroom realities. This distinction underscores the complementary nature of pre-service and in-service perspectives and highlights the necessity for future research that bridges these two groups.

It is recommended that AIL be incorporated into teacher education curricula. Specifically, technology literacy should be approached not merely as a technical skill but should include modules that integrate critical thinking, ethical decision-making, and data awareness [30]. Within education faculties, it is essential to introduce mandatory or elective courses that elucidate the operational processes, ethical dimensions, and pedagogical potential of AI, particularly within Elementary and Secondary Education (ESE, SSE) and Technology and Learning Technologies (TLT) programs [11]. Given that AIL transcends technical knowledge and necessitates critical thinking and digital ethical awareness, it should be supported by interdisciplinary content [43,53]. To ensure that GenAI tools encompass creative, effective, and pedagogical content, scenario-based, application-focused activities and online training should be organized [7,21,50].

From a curriculum design perspective, the limited prevalence of formal AI training necessitates incorporating AI literacy into teacher education programs. AI-related competencies should be integrated into core coursework, including educational technology, pedagogy, and assessment. Research shows that early, curriculum-embedded technology education is more effective than isolated interventions in fostering sustainable pedagogical practices [60,61]. Integrating GenAI within existing courses may facilitate both technical competencies and critical perspectives on AI use.

This finding indicates the need for clearer institutional and national guidelines on AI integration in teacher education. International frameworks emphasize AI literacy as a fundamental component of educators' professional competence, highlighting ethics, transparency, and human-centered use [24]. Without policy-level standards for AI competencies, responsibility for AI training remains uneven across institutions, potentially exacerbating disparities in teacher preparedness [69]. Establishing minimum expectations for AI-related skills within teacher education policies may ensure more systematic preparation. It is recommended that AI training be disseminated and standardized prior to service, as the study indicates that teacher candidates from various disciplines who received AI training exhibited significantly higher literacy levels. Training content should be diversified to encompass ethical considerations, data security, and classroom application [2,51].

Furthermore, it is recommended that distinct AI approaches be developed for different disciplines. The inclination of departments with a focus on science and technology towards AI should not disadvantage other departments. To mitigate this disparity, AI usage scenarios specific to fields such as English, Turkish, or special education should be developed, and common intersections with subject teaching content should be established [1]. Developing digital strategies for gender equality is also recommended. The observation that male teacher candidates have higher AIL scores than their female counterparts' highlights gender-based disparities. Female teacher candidates should be encouraged to engage with technology, and mentoring mechanisms should be developed [24,31]. It is further recommended that opportunities be provided for diversifying and experimenting with GenAI applications. The study found that teacher candidates who engaged with a variety of GenAI applications demonstrated higher acceptance and AIL scores. Therefore, candidate teachers should be afforded environments where they can experience multiple GenAI tools, and the conscious and purposeful use of these applications in teaching processes should be supported [7,51] It is advised that the gap between GenAI acceptance and AIL be minimized. For teacher candidates with high acceptance levels but low knowledge levels, content and applications aimed at understanding the background of these tools should be provided, rather than focusing solely on the use of GenAI tools. A balanced development of technical, social, and pedagogical awareness regarding GenAI is essential [22,35,59]. It is recommended that training on the use of GenAI tools be made mandatory both in-service and prior to employment. The study reveals that teacher candidates who receive GenAI training achieve higher literacy levels. Consequently, teacher training programs should include courses that cover the use, objectives, and content structure of GenAI [43].

## Author contributions

**Conceptualization:** Gözdegül Arık Karamık.

**Data curation:** Gözdegül Arık Karamık, Ali Özkaya.

**Formal analysis:** Gözdegül Arık Karamık.

**Funding acquisition:** Gözdegül Arık Karamık.

**Investigation:** Berker Kurt, Gözdegül Arık Karamık.

**Methodology:** Berker Kurt, Gözdegül Arık Karamık, Ali Özkaya.

**Project administration:** Gözdegül Arık Karamık.

**Resources:** Gözdegül Arık Karamık.

**Software:** Gözdegül Arık Karamık.

**Supervision:** Gözdegül Arık Karamık.

**Validation:** Gözdegül Arık Karamık.

**Visualization:** Gözdegül Arık Karamık.

**Writing – original draft:** Berker Kurt, Gözdegül Arık Karamık, Ali Özkaya.

**Writing – review & editing:** Gözdegül Arık Karamık.

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
