## [Decision Letter · Decision Letter 0]

9 Dec 2025

Dear Dr. Karamık,

We note that some of the comments of the reviewers refer to specific articles for you to cite. Please note that it is not mandatory that you cite these specific articles and you are welcome to seek alternatives manuscripts in the literature that are relevant to your manuscript’s content.

We look forward to receiving your revised manuscript.

Kind regards,

Andrea Cioffi

Academic Editor

PLOS One

Journal Requirements:

2. In the online submission form you indicate that your data is not available for proprietary reasons and have provided a contact point for accessing this data. Please note that your current contact point is a co-author on this manuscript. According to our Data Policy, the contact point must not be an author on the manuscript and must be an institutional contact, ideally not an individual. Please revise your data statement to a non-author institutional point of contact, such as a data access or ethics committee, and send this to us via return email. Please also include contact information for the third party organization, and please include the full citation of where the data can be found.

3. Please ensure that you include a title page within your main document. You should list all authors and all affiliations as per our author instructions and clearly indicate the corresponding author

Reviewers' comments:

Reviewer's Responses to Questions

**Comments to the Author**

1. Is the manuscript technically sound, and do the data support the conclusions?

Reviewer #1: Yes

Reviewer #2: Yes

2. Has the statistical analysis been performed appropriately and rigorously?

Reviewer #1: Yes

Reviewer #2: Yes

3. Have the authors made all data underlying the findings in their manuscript fully available?

Reviewer #1: Yes

Reviewer #2: Yes

4. Is the manuscript presented in an intelligible fashion and written in standard English?

Reviewer #1: Yes

Reviewer #2: Yes

Reviewer #1: The manuscript addresses an important and timely topic: the relationship between pre-service teachers’ acceptance of generative AI (GAI) and their AI literacy (AIL). The mixed-methods design, large sample size, and clear instruments are commendable. However, the paper requires substantial revision before it can be considered for publication. The following points highlight areas that need improvement.

a) The manuscript inconsistently refers to “General Artificial Intelligence (GAI)” instead of “Generative AI.” This is potentially misleading, as “general AI” has a different meaning in the literature. The terminology should be standardized throughout the paper.

b) The authors should explicitly define “AI literacy” and situate it within established frameworks (e.g., Long & Magerko, 2020; UNESCO, 2021).

c) The exclusive focus on pre-service teachers is a limitation. While their perspectives are valuable, they are not yet fully immersed in the realities of classroom practice. The authors should acknowledge this limitation and discuss how findings may differ from those of in-service teachers.

d) To strengthen the discussion, the authors should integrate literature on faculty and in-service teacher adoption of generative AI, which provides a more grounded perspective. Relevant studies include:

◦ Miranda-González, F. J., & Chamorro-Mera, A. (2025). Exploring the adoption of generative artificial intelligence tools among university teachers. Higher Education Research & Development. https://doi.org/10.1080/07294360.2025.2559648

◦ Chamorro-Mera, A., & Miranda-González, F. J. (2025). Gen-AI tools in academia: A cluster analysis of university faculty adoption. Multidisciplinary Journal for Education, Social and Technological Sciences, 12(2). https://doi.org/10.4995/muse.2025.23908

◦ Almisad, B., & Aleidan, A. (2025). Faculty perspectives on generative artificial intelligence: Insights into awareness, benefits, concerns, and uses. Frontiers in Education, 10, Article 1632742. https://doi.org/10.3389/feduc.2025.1632742

◦ Singh, S., & Strzelecki, A. (2025). Academics as adopters of generative AI: An application of diffusion of innovations theory. Education and Information Technologies, 30(11), 20495–20522. https://doi.org/10.1007/s10639-025-13835-8

e) The description of the mixed-methods design is clear, but the rationale for choosing pre-service teachers over in-service teachers should be elaborated.

f) The qualitative sample (48 participants) is relatively small compared to the quantitative sample. The authors should justify how saturation was achieved and how representativeness was ensured.

g) The data availability statement is inconsistent: one section claims full availability, while another restricts access due to personal data. This contradiction must be resolved.

h) The findings are presented clearly, but the discussion could be more strongly tied to theoretical frameworks such as the Technology Acceptance Model (TAM) or Diffusion of Innovations Theory.

i) The authors should expand on the implications of the very low percentage of participants (6.1%) who received formal AI training. This is a critical finding that deserves more emphasis in terms of curriculum design and policy recommendations.

j) The abstract and introduction are overly verbose. Shorter, more concise sentences would improve readability.

k) The discussion should include a stronger reflection on ethical concerns (bias, privacy, transparency) and how these intersect with teacher education.

Reviewer #2: Include Recent and Relevant Literature

The manuscript would benefit from citing two recent studies directly aligned with generative AI awareness and the pedagogical use of AI in education. I strongly recommend adding the following references to strengthen the theoretical framework and discussion:

Semerci Şahin et al. (2025) – development of the Generative AI Awareness Scale for secondary school students, which directly relates to AI literacy and awareness measurement approaches.

Reference:

Semerci Şahin, R., Özbay, Ö., Çınar Özbay, S., & Durmuş Sarıkahya, S. (2025). Development of the generative artificial intelligence awareness scale for secondary school students in Türkiye. European Journal of Pediatrics, 184(9), 585. https://doi.org/10.1007/s00431-025-06435-8

Sarıkahya et al. (2025) – a qualitative study examining faculty members' experiences with ChatGPT in nursing education, highly relevant for understanding generative AI acceptance, pedagogical challenges, and literacy gaps.

Reference:

Sarıkahya, S. D., Özbay, Ö., Torpuş, K., Usta, G., & Özbay, S. Ç. (2025). The impact of ChatGPT on nursing education: A qualitative study based on the experiences of faculty members. Nurse Education Today, 152, 106755. https://doi.org/10.1016/j.nedt.2025.106755

These studies will specifically enrich the authors’ sections on GAI acceptance, literacy, and discipline-based differences in attitudes toward AI.

Introduction – Reduce Repetition

Some sentences repeat similar ideas about AI's growing role in education. A brief refinement would improve clarity.

Conceptual Clarification

The manuscript alternates between “GAI” and “General Artificial Intelligence.” Please ensure consistent terminology throughout.

Figures 4 and 5

The thematic maps are informative but somewhat difficult to read due to resolution. Increasing clarity or resolution would strengthen presentation.

Discussion Section

The discussion is thorough; however, condensing overlapping ideas and explicitly linking the results to the newly suggested studies would improve coherence.

With these minor revisions addressed particularly the inclusion of the two recent and relevant references the manuscript will be suitable for publication.

**Do you want your identity to be public for this peer review?** For information about this choice, including consent withdrawal, please see our Privacy Policy

Reviewer #1: **Yes:** F. Javier Miranda

Reviewer #2: No

---

## [Author Response · Author response to Decision Letter 1]

5 Jan 2026

The corrections made in the article have been uploaded to the system under the heading “Response to Reviewers”.

---

## [Decision Letter · Decision Letter 1]

15 Jan 2026

Dear Dr. Karamık,

plosone@plos.org . A letter that responds to each point raised by the academic editor and reviewer(s). You should upload this letter as a separate file labeled 'Response to Reviewers'.A marked-up copy of your manuscript that highlights changes made to the original version. You should upload this as a separate file labeled 'Revised Manuscript with Track Changes'.An unmarked version of your revised paper without tracked changes. You should upload this as a separate file labeled 'Manuscript'.

We look forward to receiving your revised manuscript.

Kind regards,

Andrea Cioffi

Academic Editor

PLOS One

**Journal Requirements:**

**Additional Editor Comments:**

Please, also address a small number of minor points to further strengthen transparency: (i) revise the Data Availability Statement to provide a clear, non-discretionary procedure for accessing the underlying dataset (and clarify whether a de-identified dataset can be shared); (ii) clarify the wording and measurement units for GenAI usage categories (as the current distribution may be open to misinterpretation); (iii) complement the main statistical comparisons with appropriate effect sizes (and, where feasible, confidence intervals); and (iv) slightly improve the integration of quantitative and qualitative findings by briefly stating how survey results informed the interview guide and by linking selected quotes to key quantitative patterns.

Reviewers' comments:

Reviewer's Responses to Questions

**Comments to the Author**

Reviewer #1: All comments have been addressed

2. Is the manuscript technically sound, and do the data support the conclusions?

Reviewer #1: Yes

3. Has the statistical analysis been performed appropriately and rigorously?

Reviewer #1: Yes

4. Have the authors made all data underlying the findings in their manuscript fully available?

Reviewer #1: Yes

5. Is the manuscript presented in an intelligible fashion and written in standard English?

Reviewer #1: Yes

Reviewer #1: I appreciate the revisions made to the manuscript. The authors have adequately addressed all the comments raised in the previous review round, and the changes introduced have improved the clarity and overall quality of the paper. I do not identify any remaining issues that require further modification.

In light of this, I consider the manuscript suitable for publication and recommend its acceptance.

**Do you want your identity to be public for this peer review?** For information about this choice, including consent withdrawal, please see our Privacy Policy

Reviewer #1: **Yes:** F. Javier Miranda

---

## [Author Response · Author response to Decision Letter 2]

22 Jan 2026

We thank Reviewer 1 for his/her comment: “I appreciate the revisions made to the manuscript. The authors have adequately addressed all the comments raised in the previous review round, and the changes introduced have improved the clarity and overall quality of the paper. I do not identify any remaining issues that require further modification. In light of this, I consider the manuscript suitable for publication and recommend its acceptance.” There have been no corrections from the reviewers regarding the second revision. As the authors, we have revised our manuscript in light of the “additional editor comments” and uploaded the 'Revised Manuscript with Track Changes' file.

Additional Editor Comments:

(i) Revise the Data Availability Statement to provide a clear, non-discretionary procedure for accessing the underlying dataset (and clarify whether a de-identified dataset can be shared

R (i): In order to preserve author anonymity during the review process, the initial version of the manuscript referred only to the platform where the data were deposited. Following this comment, the Data Availability Statement has been revised, and the DOI corresponding to the dataset has now been explicitly included in the manuscript to ensure transparent and non-discretionary access to the underlying data.

(ii) clarify the wording and measurement units for GenAI usage categories (as the current distribution may be open to misinterpretation

R (ii): We appreciate your comment. We wish to clarify that the usage of GenAI is operationalized and reported in Table 1 as Daily GenAI Usage (hours/day), which explicitly denotes the measurement unit employed for categorizing participants' GenAI use. We believe this phrasing minimizes potential ambiguity in interpreting the distribution. We understand from your comment that the primary concern pertains to the clarity of wording and measurement units; however, we are willing to make further revisions should you deem additional clarification or modifications necessary.

(iii) complement the main statistical comparisons with appropriate effect sizes (and, where feasible, confidence intervals);

R (iii): The desired statistical results have been added to the article.

(iv) slightly improve the integration of quantitative and qualitative findings by briefly stating how survey results informed the interview guide and by linking selected quotes to key quantitative patterns.

R (iv): “The results indicated that participants' levels of acceptance of GenAI and AIL varied according to their departmental affiliation. To explore the potential reasons for these departmental differences in greater depth, qualitative interview instruments were developed.” added the article.

---

## [Editor Report · Decision Letter 2]

29 Jan 2026

Investigating the Correlation Between Candidate Teachers' Acceptance of Generative Artificial Intelligence and Artificial Intelligence Literacy Across Various Disciplines

PONE-D-25-59540R2

Dear Dr. Karamık,

We’re pleased to inform you that your manuscript has been judged scientifically suitable for publication and will be formally accepted for publication once it meets all outstanding technical requirements.

Kind regards,

Andrea Cioffi

Academic Editor

PLOS One
---

## [Editor Report · Acceptance letter]

PONE-D-25-59540R2

PLOS One

Dear Dr. Karamık,

I'm pleased to inform you that your manuscript has been deemed suitable for publication in PLOS One. Congratulations! Your manuscript is now being handed over to our production team.

Kind regards,

on behalf of

Dr. Andrea Cioffi

Academic Editor

PLOS One